# Nanocapsules of ZnO Nanorods and Geraniol as a Novel Mean for the Effective Control of *Botrytis cinerea* in Tomato and Cucumber Plants

**DOI:** 10.3390/plants12051074

**Published:** 2023-02-28

**Authors:** Panagiota Tryfon, Nathalie N. Kamou, Akrivi Pavlou, Stefanos Mourdikoudis, Urania Menkissoglu-Spiroudi, Catherine Dendrinou-Samara

**Affiliations:** 1Laboratory of Inorganic Chemistry, Department of Chemistry, Aristotle University of Thessaloniki, 54124 Thessaloniki, Greece; 2Pesticide Science Laboratory, School of Agriculture, Faculty of Agriculture Forestry and Natural Environment, Aristotle University of Thessaloniki, 54124 Thessaloniki, Greece; 3Biophysics Group, Department of Physics and Astronomy, University College London, London WC1E 6BT, UK; 4UCL Healthcare Biomagnetics and Nanomaterials Laboratories, 21 Albemarle Street, London W1S 4BS, UK

**Keywords:** nanoformulations, nanofungicides, essential oil encapsulation, antifungal activity, geraniol control release, plant protection

## Abstract

Inorganic-based nanoparticle formulations of bioactive compounds are a promising nanoscale application that allow agrochemicals to be entrapped and/or encapsulated, enabling gradual and targeted delivery of their active ingredients. In this context, hydrophobic ZnO@OAm nanorods (NRs) were firstly synthesized and characterized via physicochemical techniques and then encapsulated within the biodegradable and biocompatible sodium dodecyl sulfate (SDS), either separately (ZnO NCs) or in combination with geraniol in the effective ratios of 1:1 (ZnOGer1 NCs), 1:2 (ZnOGer2 NCs), and 1:3 (ZnOGer2 NCs), respectively. The mean hydrodynamic size, polydispersity index (PDI), and ζ-potential of the nanocapsules were determined at different pH values. The efficiency of encapsulation (EE, %) and loading capacity (LC, %) of NCs were also determined. Pharmacokinetics of ZnOGer1 NCs and ZnOGer2 NCs showed a sustainable release profile of geraniol over 96 h and a higher stability at 25 ± 0.5 °C rather than at 35 ± 0.5 °C. ZnOGer1 NCs, ZnOGer2 NCs and ZnO NCs were evaluated in vitro against *B. cinerea,* and EC_50_ values were calculated at 176 μg/mL, 150 μg/mL, and > 500 μg/mL, respectively. Subsequently, ZnOGer1 NCs and ZnOGer2 NCs were tested by foliar application on *B. cinerea*-inoculated tomato and cucumber plants, showing a significant reduction of disease severity. The foliar application of both NCs resulted in more effective inhibition of the pathogen in the infected cucumber plants as compared to the treatment with the chemical fungicide Luna Sensation SC. In contrast, tomato plants treated with ZnOGer2 NCs demonstrated a better inhibition of the disease as compared to the treatment with ZnOGer1 NCs and Luna. None of the treatments caused phytotoxic effects. These results support the potential for the use of the specific NCs as plant protection agents against *B. cinerea* in agriculture as an effective alternative to synthetic fungicides.

## 1. Introduction

*Botrytis cinerea* (*B. cinerea*) is an essential necrotrophic fungal pathogen with a broad host range. It is a worldwide fungus that causes grey mold disease with high damage in more than 500 plant species [1]. The economic losses associated with *B. cinerea* are a border in the agricultural industry, especially regarding cucumbers and tomatoes, which belong to the world’s most economically valuable and nutritional vegetable crops. Moreover, cucumber is highly susceptible to necrotrophic fungal pathogens, *B. cinerea* being the most common [2]. Tomato (*Solanum lycopersicum*) is the second most important crop worldwide, covering 5 Mha of arable land and producing more than 180 million tons of product (FAOSTAT). Additionally, Europe’s estimated value of total production for 2020 was more than 16.5 million EUR [3]. Panno et al. (2021) list the number of pathogens that affect tomato production and the disease caused by *B. cinerea* as one of the most important reasons for yield reduction [4]. These losses are observed both in the open field and under greenhouse conditions, where the effect of the pathogen on these hosts is even more disastrous [5]. Synthetic compounds are used to manage plant diseases, even though they frequently degrade slowly in nature, are applied in high concentrations, and need repeat application [6]. The reckless use of synthetic chemical fungicides has led to the appearance of resistant isolates, and it has prompted a search for new and effective fungicide agents, such as natural bioactive compounds.

The interest in biological activity of natural compounds has been growing fast, and a considerably high number of applications are explored and reported in recent studies. Specifically, essential oils (EOs) or their constituents exhibit intriguing antibacterial, anti-fungal, insecticidal, nematicidal, repellant, or antiphytoviral activity [7,8,9]. In plant protection, they are considered as promising a safe means for reducing the harmful effects of synthetic chemical pesticides, and commercial products have already been granted in the market (EU pesticide database). Geraniol, the main constituent of many EOs, has been reported to possess powerful antimicrobial, nematicidal, and insecticidal properties in plant disease management [7,8,9]. These compounds, in addition to their ability to control plant infections, also prevent the deterioration of ecological balance. Despite their many advantages, EOs have significant drawbacks, including high cost, physical and chemical instability, low water solubility, oxygen breakdown, phytotoxicity, and high vapor pressure. Furthermore, they have organoleptic effects, which result in unappealing flavors and odors for customers in products containing them [10]. In this case, proper formulation is needed, and nanotechnology demonstrates great promise in improving pest management strategies for protecting and retaining active ingredients to extend their shelf life and effectiveness.

Encapsulation is an appropriate method for overcoming issues caused by the instability and volatility of the EOs and protecting active ingredients as a means of improving their distribution and controlled release [11]. Surfactant-type molecules are required to form nanocapsules (NCs). The polyions, such as sodium dodecyl sulfate (SDS), have a crucial role in the release of active ingredients in biological applications while they are also biocompatible and biodegradable [12].

The antifungal activity of inorganic-based nanoparticles as well as of EOs has been evaluated against a variety of phytopathogenic fungi. Geraniol has shown beneficial antifungal effects against *Drechslera oryzae*, *Fusarium oxysporum*, *Rhizoctonia solani*, *Sclerotium rolfsii* [13], and *B. cinerea* [9]. Meanwhile, ZnO NPs exhibit considerable advantages over inorganic oxides owing to their low cost as well as their proven antimicrobial and antifungal activity, while they are also biocompatible and environmentally friendly at low concentrations [14]. Zn is an essential micronutrient that enhances enzyme reactions, improves the effectiveness of photosynthesis, and enhances the antioxidant system in tomato plants [15]. The US Food and Drug Administration has recognized ZnO as a “generally recognized as safe (GRAS)” substance [16]. Size- and shape-dependent antifungal activity is being carried out, too. Specifically, spherical ZnO nanoparticles (crystallite size 14 nm) were found to exhibit better antifungal activity against *Fusarium solani* (reducing growth by up to 65%) than ZnO platelets and ZnO hexagonal rods with a length > 1 μm [17]. Meanwhile, the inhibitory effect of pegylated ZnO@PEG Nanoflowers (40 nm) on *S. sclerotiorum*, *B. cinerea* [18], and ZnO (70 ± 15 nm) on *Penicillium expansum* [19] was found to be moderate. The antifungal activity of ZnO Nanorods (ZnO NRs), though, is less studied. There is evidence showing that rod coarser with almost needle-like sharp ends differs from spherical shapes and contributes to the more significant mechanical damage of the cell membrane, thus possessing enhanced antimicrobial and/or antibacterial activity [20].

Despite the antifungal study of separate ZnO NPs, nanocapsules of *Zataria multiflora* essential oil (ZnO-ZmEO) were investigated against six isolates of *Fusarium*. The mycelial growth inhibitory of ZnO-ZmEO was increased by 42.70% compared to the pure ZnO, and by 66.33% compared to EO [21]. This EO was also tested against *A. solani* and compared to both the initial ZnO and the reference fungicide chlorothalonil, showing NCs with the highest antifungal effect at 53.33% [22]. However, despite these promising results, relatively little is reported regarding this multifactorial matter, which consists of the method of study of the NP physicochemical properties (particle size, shape, surface charge, etc.) and the fungal strain that is targeted.

The current work aimed at preparing and characterizing hydrophilic nanocapsules that consisted of ZnO NRs either separate and/or in combination with commercial geraniol in different ratios (1:1, 1:2, and 1:3, respectively). Nanoencapsulation was assisted by sodium dodecyl sulfate (SDS) surfactant to improve their suspension stability against flocculation. In this concept, hydrophobic ZnO@OAm NRs were firstly synthesized and further physicochemical characterized. The entrapped geraniol is protected in this nano-delivery system. Furthermore, pharmacokinetic study for analyzing the release mechanism was performed. The antifungal ability of the effective NCs applied against *B. cinerea* in vitro and *in planta* on cucumber and tomato plants was also investigated in order to propose an alternative holistic strategy against this important pathogen.

## 2. Results

### 2.1. Physicochemical Characterization of ZnO@OAm NRs

The composition and the physicochemical properties of the as-synthesized ZnO@OAm NRs were characterized using different methods. Initially, an X-ray diffraction (XRD) analysis was performed to evaluate the crystalline structure and phase purity of the synthesized ZnO@OAm NRs to confirm their ability to be used further in the form of NCs. The XRD pattern of ZnO@OAm NRs (Figure 1) displayed the prominent diffraction peaks located at 31.7°, 34.4°, and 36.2° that were indexed to the hexagonal wurtzite phase (JCPDS card #89-0510, space group P63mc (186)), and they corresponded to the Miller Indices (1 0 0), (0 0 2), and (1 0 1), respectively [18,23]. The average crystallite size was calculated according to Debye’s Scherer equation using the full width at half-maximum (FWHM) data to be about 18 nm. Lattice parameters were estimated at *α* = *b* = 3.2268 Ǻ and *c* = 5.2688 Ǻ. The d-spacing values were also measured for each of the three major diffraction peaks at 2.82, 2.61, and 2.48 Ǻ, respectively (Appendix A).

Thermogravimetric analysis (TGA) is an effective method to determine the thermal stability and the amount of the organic coating (% *w*/*w*) of ZnO@OAm NRs. Three steps of weight loss are presented in the TGA curve (Appendix A). Specifically, a weight loss of 0.8% *w*/*w* is due to the evaporation of the water absorbed on the surface of NRs. A rate of 10% *w*/*w* of mass loss has occurred in the temperature range 150 °C ≤ 𝑇 ≤ 714 °C due to the double layer of OAm. Meanwhile, weight loss maintains a threshold at higher temperatures (>714 °C).

The morphology of ZnO@OAm NRs was confirmed using transmission electron microscopy (TEM) analysis. The particles were rod-shaped (Figure 2A) with a mean width of 38 ± 1.21 nm (Figure 2B) and an average length of 82.5 ± 1.60 nm (Figure 2C).

The Fourier-transform infrared (FT–IR) spectrum of ZnO@OAm NRs (4000–400 cm^−1^) is shown in Appendix A. The peak at 479 cm^−1^ is associated with the characteristic Zn–O stretching vibration band that confirmed the formation of rod-shaped ZnO particles. The peaks at 2922 and 2846 cm^−1^ are ascribed to *ν_as_*(C–H) and *ν_s_*(C–H) stretching, respectively. It also displays the transmission bands at 1562 cm^−1^ (ΝH_2_) and 1451 cm^−1^ (CH_3_), and 718 cm^−1^ (–C–C–) [24].

Τhe UV-Vis absorbance spectrum of the as-prepared ZnO@OAm NRs suspension is shown in Appendix A. The broad band at 369 nm is typical for the absorption of the semiconductor. The band gap was calculated at 3.17 eV.

### 2.2. Physicochemical Characterization of ZnO-Based Nanocapsules

#### 2.2.1. Morphology of ZnO NCs

The hydrophobic ZnO@OAm NRs were encapsulated in SDS. The ZnO-based NCs were designed based on an oil-in-water template using SDS as emulsifier at critical micelle concentration (CMC, 19.5 mM). The lowest amount of SDS required to form micelles was utilized in order to provide small micelles, while hydrophobic van der Waals interactions of oleylamine on the nanoparticle surface favor further clustering. The TEM image (Figure 3) illustrates the resulted self-organized assemblies with a mean diameter of approximately 400 nm. These are irregular in shape with spiky edges.

#### 2.2.2. Encapsulation Efficiency, Loading Capacity of NCs, and Release Profile of Geraniol from NCs

Encapsulation ability of ZnO@OAm NRs with three different ratios of geraniol (1:1, 1:2, and 1:3 *w*/*w* ZnO NRs: geraniol) in the presence of SDS were recorded. The entrapped geraniol in ZnOGer1 NCs, ZnOGer2 NCs, and ZnOGer3 NCs was calculated by EE (%) at 63%, 78%, and 33%, respectively, while the LC (%) was found at 34%, 16%, and 7%, correspondingly. Based on these results, ZnOGer1 NCs and ZnOGer2 NCs were selected for further study, since the LC (%) of ZnOGer3 NCs was the lowest. The thermal stability of ZnOGer1 NCs and ZnOGer2 NCs and the release profile of geraniol from NCs formulations were investigated at two different temperatures, 25 °C and 35 °C for 96 h. Temperature is an essential factor in the case of volatile compounds in the practical development of nano-formulations in agriculture. It should be noted that the release profile assays were carried out in the same conditions as the current bioassay study of NCs against *B. cinerea*. The released profile of geraniol from the NCs was estimated via a dialysis bag approach at pH = 7.2 and based on principles of pharmacokinetics. Ionic release of zinc ions was found to be negligible.

The geraniol release profile was first presented using the zero-order model (Figure 4A) and at the applied temperatures of 25 °C and 35 °C, respectively. Initially, no significant differences were observed in the geraniol release profiles at different temperatures, with approximately 20% compound after around 3 h. However, significant differences were noticed at the cumulative geraniol release (Q%) at 4 h. For ZnOGer1, Q_4h_% = 19 and 27% release, while for ZnOGer2 NCs, Q_4h_% = 22% and 34% at 25 °C and 35 °C, respectively. It can thereby be noticed that geraniol had a more significant release as the temperature increased. After this period, the release showed a stable tendency in both ZnOGer1 NCs with Q_96h_% = 35% at 25 °C and Q_96h_% = 39% at 35 °C, and ZnOGer1 NCs with Q_96h_% = 44% at 25 °C and Q_96h_% = 49% at 35 °C.

The first-order model was also applied (Figure 4B), and according to the data shown in Appendix A, the fitting value was provided at approximately R^2^ = 0.91, verifying a concentration-dependent release profile. The Higuchi model (Figure 4C), with around R^2^ = 0.98 and K_H_ value from 9.42 to 14.52, confirmed the diffusion mechanism release. The Korsmeyer–Peppas model was determined to categorize the diffusion mechanism (Figure 4D) based on the N value (slope) (Appendix A), revealing a non-Fickian diffusion release of geraniol from ZnOGer1 NCs and a normal Fickian from ZnOGer1 NCs.

#### 2.2.3. Hydrodynamic Size, ζ-Potential, and Polydispersity

The particle size, polydispersity (PDI), and zeta potential (ζ-potential) of ZnOGer1 NCs, ZnOGer2 NCs, and ZnO NCs were measured at three different pH values (6, 7, and 8) and are summarized in Table 1 and presented in Figure 5. These values of pH were selected in order to mimic a real environment of plant growth, since soils and the stems of plants appear to have different pH. The increase in the amount of the entrapped geraniol in NCs (ZnOGer2 NCs, 1:2 ratio) resulted in a slight increment in the mean of hydrodynamic size from 144 ± 1.51 nm to 180 ± 5.31 nm, respectively, at pH = 7. Relatively bigger hydrodynamic sizes were determined for ZnO NCs (up to 256 nm at pH = 8). The hydrodynamic sizes of ZnOGer2 are slightly different in proportion to pH values but are still in the nanoregime. PDI values of all the NCs were found to be mainly in the range of 0.31–0.46, which indicates the uniformity of NCs. The charge of NCs and the resultant repulsion to each other is measured as the zeta potential. Negative ζ-potential values of ZnOGer1 NCs, ZnOGer2 NCs and ZnO NCs were found approximately −31 ± 1.29 mV, −56 ± 3.00 mV, and −53 ± 3.16 mV, respectively. The higher the repulsion, the higher the stability of NPs, and no agglomeration is observed.

### 2.3. Bioassays Results

#### 2.3.1. In Vitro Antifungal Activity

The antifungal activity of the as-prepared ZnO NCs, ZnOGer1 NCs, and ZnOGer2 NCs, and their potency against *B. cinerea* were assessed by determining the EC_50_ value (Appendix A). The difference in the mycelium diameter indicates the sensitivity of this fungus to various concentrations of NCs. EC_50_ values of ZnOGer1 NCs, ZnOGer2 NCs against *B. cinerea* were calculated at 176 µg/mL and 150 µg/mL, respectively. Their superiority was evident when compared to the ZnO NCs which demonstrated an EC_50_ value >500 μg/mL. Tested SDS showed a 5% inhibition of fungal growth.

#### 2.3.2. In Planta Antifungal Activity

The severity of the disease on cucumber and tomato plants was measured to assess the antifungal activity of ZnOGer1 NCs and ZnOGer2 NCs, since they showed higher efficacy as compared to ZnO NCs (EC_50_ > 500 μg/mL). As shown in Figure 6 and Figure 7, the study revealed that both treatments considerably reduced the disease index (DI) compared to the *B. cinerea* control treatment. Regarding the chemical control (Luna SC), in the case of cucumber, the treatments with both the NCs were significantly (*p* ≤ 0.05) more effective in inhibiting *B. cinerea* (Figure 6). When applied on tomato, the treatments of Luna SC and ZnOGer2 NCs were the most effective (Figure 7). Remarkably, both ZnOGer1 NCs, and ZnOGer2 NCs did not cause any phytotoxicity to either host after 96 h.

## 3. Discussion

The utilization of EOs and inorganic-based nanomaterials have been considered in the management of phytopathogens. One-dimensional nanostructures, such as nanorods, have attracted interest owing to their ability to disrupt the homeostasis of cells better than spherical shapes [25], thus promoting human cell lysis and apoptosis [26]. Nanoencapsulation of hydrophobic active compounds such as geraniol and ZnO NRs could improve their antifungal activity, leading to effective delivery and valid release in controlling and managing soil-borne diseases [27]. However, limited studies mention the concept of the encapsulation of geraniol. To the best of our knowledge, it is the first attempt where geraniol and ZnO NRs have been combined and encapsulated using an SDS stabilizer. Nevertheless, the current study synthesizes ZnO-based NCs (ZnO NCs, ZnOGer1 NCs, ZnOGer2 NCs, and ZnOGer3 NCs) to evaluate their stability and to test antifungal activity against an important necrotrophic fungal pathogen, *B. cinerea*.

In order to prepare ZnO-based NCs, initially, hydrophobic ZnO@OAm NRs were synthesized through a solvothermal process, an eco-friendly, low-cost, and high-yield method, in the presence of OAm. Subsequently, the physicochemical characterization of NRs was investigated via various techniques. The role of OAm on the size- and shape-controlled synthesized NRs is triple as a surfactant agent, a solvent medium, and a reducing agent [28]. The amine group is attracted to particles’ (0001) polar surfaces. This resulted in elongation in the (002) phase and the growth of rod-shaped particles. The “cis” configuration and buckled molecular chain of the OAm molecule result in an apparent rod length with limited opportunities for long-range organization [29].

The crystallite size of ZnO@OAm NRs was calculated at 18 nm, and the sharp diffraction peaks indicate the well-crystallized nanoparticles. The morphology of the rod-shaped ZnO particles was determined through the TEM image (mean width 38 ± 1.21 nm; mean length 83 ± 1.60 nm). The growth of rod-like nanostructures is explained by (i) the high surface energy of ZnO NRs, which led to their agglomeration in the absence of a high percentage of surfactant (10% *w*/*w*, Appendix A), and (ii) the elevation of the liquid’s temperature (200 °C for 8 h) caused by the solvothermal process in the Teflon-lined stainless-steel autoclave [30]. However, ZnO NRs were synthesized via a hydrothermal method in autoclave, resulting in elongated rod-shaped particles with crystallite size (>100 nm) and length (>1 μm) [17,31] rather than ZnO@OAm NRs (18 nm; length 83 ± 1.60 nm). The increased particle sizes are attributed to the addition of water and/or ethanol that generates high pressure in the process and thereby modifies the nucleation rate of the ZnO crystals, which is in contrast with the present smaller particles that were prepared in the sole use of OAm.

Light conditions are a factor that may affect the efficient delivery of foliar Zn-based NCs to plants. Under growth lights, the gas exchange at the stomata and the transport of NCs are activated upon foliar application. ZnO is an n-type semiconductor with photocatalytic properties, and its antifungal action is activated under UV–Vis irradiation. In the current study, the *in planta* experiments on plants were carried out under greenhouse conditions (130 ± 20 μmol quanta m^−2^ s^−1^). For this reason, optical properties of ZnO@OAm NRs were determined via UV–Vis spectroscopy. The absorbance peak at 369 nm is a characteristic of hexagonal wurtzite ZnO structure [30], which was also certified by TEM and XRD techniques. In the case of nanomaterials, the relationship between particle size and band gap is inversely proportional. The calculated band gap (3.17 eV) [32] confirmed the particles in the nanoscale, compared with the band gap of single-side ZnO (3.3 eV) [33]. The band gap shrinks as the number of NPs increases, although it never reaches zero [34]. A decreased NP size and a more significant density of defects in the nanomaterial [30,35] is linked with the antimicrobial impact of NPs [36].

The EE (%) of geraniol in ZnOGer1 NCs, ZnOGer2 NCs and ZnOGer3 NCs were calculated at 63%, 78%, and 33%, respectively, and the LC (%) at 34%, 16%, and 7%, respectively. In our previous research, the EE% of geraniol in a formulation of geraniol oil in water (O/W) nanoemulsions (GNEs) was calculated at 57%, while the LC was at 14% [9]. Similarly, Yegin et al. (2016) showed the EE for geraniol in Pluronic F-127 NPs (size 412 nm) at 57.5% [37]. Thus, ZnO-based NCs improve encapsulation ability.

Particle size significantly impacts the functional characteristics of capsules and influences the interaction of NCs with plant leaves [38]. Nanoencapsulation uses a carrier with a diameter of less than 1 micron (1000 nm) and properties distinct from conventional encapsulation [39]. The hydrodynamic size of the NCs slightly varied from 141 to 256 nm (Table 1) but are under suitable range for an application in agriculture [40]. However, based on the physicochemical characteristics of each system, hydrodynamic sizes varied as for ZnO-encapsulated with *A. sieberi* and *Z. multiflora*, EOs were found at 425 nm [22] and 157 nm, respectively. Regarding the influence of pH variations on the stability of NCs, an increase in the pH from six to eight slightly increases the size (d.nm) and PDI in contrast with the ζ-potential value. Experimental pH is of interest because of its ability to influence the size and the shape of particles, which will subsequently be implemented in bioassays. The PDI values of NCs (0.31–0.42) indicate satisfying size distribution [41]. Generally, a low PDI value indicates an efficient formulation of the system and homogeneity, which may help attain excellent stability [42].

The ζ-potential is also an important parameter to reflect the physicochemical and biological stabilities of particles in dispersion, which help the formulation to enhance long-term stability [43]. In all cases, the NCs are negatively charged due to SDS molecules [44]. Specifically, SDS is a detergent with a long aliphatic chain and a sulfate group that forms a powerful negatively charged complex (the negative charge arising from the SO_4_^2−^ groups of SDS) [45].

The pharmacokinetics of ZnOGer1 NCs and ZnOGer2 NCs showed that even after 48 h incubation at 35 °C, both NCs demonstrated a satisfactory sustainable release trend. A different release profile was observed in our previous study, and according to the zero-order model, native geraniol reached a release over 55% in 4 h approving a short-release tendency, while the same value showed GNEs after 73 h [9]. ZnOGer1 with Q_4h_% = 19% and ZnOGer2 NCs with Q_4h_% = 22% at 25 °C illustrated a more stable release profile of geraniol than GNEs with Q_4h_% = 32% due to the presence of the inorganic nanoparticles that stabilize further geraniol molecules through van der Walls interaction.

This evidence further confirms that native geraniol usually occurs with burst dissipation, resulting in short-term persistence and significantly compromising its antifungal activity [46]. Controlled release profiles can be used to achieve several benefits, including a sustained constant concentration of active compounds at the target site, predictable and reproducible release rates over a long time, protection of bioactive compounds with a short life, elimination of side effects as well as the abolition of chemical pesticides and their frequent dosing, optimized pest control, and, finally, a reduction in pesticide use [47]. SDS is widely used as an agro-surfactant that helps to deliver active ingredients and reduces surface tension to improve the absorption of NCs into leaves [48].

Study of the antifungal activity of ZnO NCs, ZnOGer1 NCs, and ZnOGer2 NCs against *B. cinerea* showed that ZnOGer2 NCs were more effective with EC_50_ = 150 µg/mL than ZnOGer1 NCs (EC_50_ = 176 µg/mL) and ZnO NCs (EC_50_ > 500 µg/mL), while the EC_50_ value of pure geraniol was found in our previous study to be equal to 235 µg/mL [9]. However, geraniol-loaded nanoemulsions (GNEs) against *B. cinerea* showed a relatively higher growth inhibition (EC_50_ = 105 μg/mL) [9], which may be due to the acute release of geraniol (Q_4h_% = 22%).

The severity of the disease on infected cucumber plants after foliar application of ZnOGer1 NCs and ZnOGer2 NCs was lower when compared to treatments with the commercial fungicide Luna. In the case of infected tomato plants, ZnOGer2 NCs and Luna showed a lower DI. Neither ZnOGer1 NCs nor ZnOGer2 NCs cause phytotoxic effects on plants. Other studies showed that the EOs of various plants were tested (0.125, 0.25, 0.5, and 1 μL/mL) against cucumber (*Cucumis sativus* L.) and tomato (*Solanum lycopersicum* L.), and exhibited a dose dependent phytotoxic activity [49]. The cultivated cucumber was more resistant to tomato at all tested EOs, similar to NCs in our research.

A number of studies on encapsulated geraniol against food-borne pathogens are reported. In 2016, Yegin et al. investigated the inhibition of *Salmonella enterica* and *Escherichia coli* O157:H7 in vitro on spinach surfaces by geraniol loaded-pluronic F127 nanoparticles [37]. For both pathogens, a decrease in the MIC of geraniol following nano-encapsulation was reported. The type of NP application onto spinach inoculated with pathogens was investigated and results showed that the immersion was more effective than the spraying technique due to the high contact between pathogen cells attached to spinach surfaces and EO-loaded NP. This finding was attributed to the fact that encapsulation of geraniol enhances their bioavailability and transport to targeted cells (*Escherichia coli* O157:H7 cells) [37].

Zn defense-related mechanisms in plants greatly vary. The outcomes of plant–pest/pathogen interactions differ, depending on the effectivity of the Zn-related responses in limiting the invader attacks as well as on the enemy’s ability to circumvent the plant defenses, in addition to other environmental conditions that can favor either the host or the invader [50]. On the other side, geraniol is able to change the hyphal morphology and cause hyphal aggregates, resulting in reduced diameters and lyses of the hyphal wall. Geraniol is also lipophilic, enabling it to interact with the cell membrane of fungus cells [51]. The encapsulation of geraniol improved its antifungal activity against *B. cinerea*, as demonstrated by the reduced EC_50_ values. The enhancement in antifungal activity was explained by the sustained release, the improved hydrophilicity, and the better penetration that resulted from the small size. Similarly, the DI were minimized upon NCs treatment, as previously reported in a similar study [22].

Surface chemistry and plant species are two significant factors with impact on the uptake and distribution of NPs. Negatively-charged NPs remained primarily adhered to the positive charged leaves via electrostatics. This correlates with higher transpiration rates and water uptake. Despite the variable plant morphology, surface charge allows the uptake and translocation of NPs from leaf to root [18,52,53]. NCs showed different results after its foliar application on tomato and cucumber leaves, possibly due to distinct surface biotransformation *in planta* (e.g., corona formation and heteroaggregation) and/or differential potential for the membrane crossing provided by the charge type and/or density. However, further research is needed to determine the way NCs interact with leaf surface and distribution. Overall, a function of particle surface chemistry mediate NCs as smart delivery systems for agrochemicals to different plants.

Regarding the interaction between NCs and leaf cells, the average radii of the pores in waxy hydrophobic cuticles are reported to be smaller than 2.4 nm, restricting the transport of particles with a larger size. However, the cuticle composition and NC surface properties may affect the efficiency of cuticular NCs uptake. The stomatal pathway is the main route for plant foliar uptake of NCs. Stomata are important for gas exchange during photosynthesis and their length and width are usually at the scale of micrometers, with variations among plant species. According to Liu et al. (2014) [54] as well as Kardiman and Raebild (2018) [55], the pore area of stomata or stomatal size in the open state is typically a few hundred square micrometers, which is sufficient for NPs to pass through. There is less awareness of the relationship between NPs and stomata since much research has yet to be performed on how NPs affect stomatal characteristics. Generally, a leaf’s abaxial and adaxial sides have differing stomatal densities [56]. Another major factor is the size of the sieve plate pores in the phloem, which in tomato pedicels is up to 600 nm [57]. The particle size distribution of ZnOGer NCs and ZnO NCs is determined approximately under 256 nm, indicating the possibility of NC entry.

To summarize, ZnO@NRs and geraniol were successfully encapsulated preparing ZnO-based NCs, and their antifungal activity against *B. cinerea* is related to various factors, as mentioned above. This mode depends on the size and surface charge of NCs, and the physiology of tomatoes and cucumbers. The effective use of these NRs was shown in this study and could be an important step towards the Green Deal, since their promising application as alternative biopesticides could result in the reduction of conventional fungicides in those two important crops.

## 4. Materials and Methods

### 4.1. Materials

#### Chemicals and Reagents

All chemicals and reagents were of analytical grade and used without any further purification: zinc acetylacetonate hydrate, Zn(acac)_2_ (Sigma-Aldrich, St. Louis, MO, USA, *M* = 263.61 g/mol), Oleylamine (C_18_H_35_NH_2_), OAm (Merck, Rahway, NJ, USA, *M* = 267.493 g/mol), Sodium Dodecyl Sulfate (CH_3_(CH_2_)_11_SO_4_Na), SDS (Sigma-Aldrich, ≥99%, *M* = 288.38 g/mol), triethylene glycol, TrEG (Merck, ≥99%, *M* = 150.17 g/mol), diethyl ether (Merck, ≥99%, *M* = 74.12 g/mol), geraniol, trans-3,7-Dimethyl-2,6-octadien-1-ol (Sigma-Aldrich, 98%, *M* = 154.25 g/mol), Milli-Q water, phosphate buffer solution, PBS (Gibco by Life Technologies, Carlsbad, CA, USA, pH 7.2, 10X), potato dextrose agar, PDA (BD Difco, Franklin Lakes, NJ, USA, Luna^®^ Sensation SC Fungicide (Bayer Crop Science, Leverkusen, Germany, fluopyram 250 g/L and trifloxystrobin 250 g/L), and methanol (SD-Fine).

### 4.2. Synthesis of ZnO@OAm Nanorods

The solvothermal method is mainly used for directing the formations of various metal oxides, such as ZnO NRs since it is an eco-friendly, high-yield, and simple process [58]. ZnO@OAm NRs were prepared by the solvothermal method: Zn(acac)_2_ (1.06 mmol) was mixed and dissolved in 4 mL of TrEG and 4 mL of OAm, under stirring at 30 °C for 15 min. The resulting solution was centrifuged into a Teflon-lined stainless-steel autoclave under a solvothermal polyol process. The reaction was carried out at 200 °C for 8 h. After the solvothermal polyol process, the solution was centrifugation at 5000 rpm for 20 min. The supernatants were discarded, and a white-bronze precipitate was acquired and washed with disolol to remove the untreated precursors. Reaction yield was calculated at 38% based on the metal precursor.

### 4.3. Preparation ZnO-Based Nanocapsules

The combined ZnO-based NCs were prepared according to the following approach. Briefly, 15 mg of as-prepared hydrophobic ZnO@OAm NRs was dissolved in diethyl ether (3 mL) and mixed with pure commercial geraniol. After that, the mixture was added to ddH_2_O solution (30 mL) with SDS (19.5 mM) and was emulsified by sonication treatment in a close vial under stable conditions (<30 °C). Subsequently, the vial was opened, and diethyl ether was evaporated to be slowly removed and to isolate nanocapsules: (i) ZnOGer1 NCs (ZnO@OAm NRs: geraniol mass ratio 1:1), (ii) ZnOGer2 NCs (ZnO@OAm NRs: geraniol mass ratio 1:2), and (iii) ZnOGer3 NCs (ZnO@OAm NRs: geraniol mass ratio 1:3). The same procedure but without the addition of geraniol was followed to prepare ZnO NCs. The prepared ZnO-based NCs were stored at room temperature (25 °C) for physicochemical characterization.

### 4.4. Physicochemical Characterization of Nanorods and ZnO-Based Nanocapsules

The synthesized nanoparticles and nanocapsules were characterized via various analytical instruments for measuring size, morphology, and stability. The average crystalline size and crystalline structure of NRs were identified by XRD using a two-cycle Rigaku Ultima+ X-ray diffractometer (Rigaku Corporation, Shibuya-Ku, Tokyo, Japan) with a Cu-Ka radiation (λ = 1.541 Å) operating at 40 kV/30 mA of the Debye–Scherrer equation. A Nicolet series FT–IR spectrometer (Nicolet iS20, Thermo Fisher Scientific, Waltham, MA, USA) with a monolithic diamond ATR crystal was used to acquire the FT-IR spectrum (4000–450 cm^−1^) of ZnO@OAm NRs. The morphology and the particle size were investigated by TEM with a JEOL JEM 1200–EX transmission electron microscope at an acceleration voltage of 120 kV. For TEM imaging, suspensions of the NPs were used, drop-casted on carbon-coated copper grids. TGA were utilized to examine the thermal stability and the amount of organic coating of NRs using SETA–RAM SetSys-1200 at a heating rate from 30 °C to 800 °C (10 °C min^−1^) under an N_2_ atmosphere. The optical properties were calculated by using a UV-Vis spectrophotometer (V-750, Jasco, Tokyo, Japan) and the band gap of NRs based on Tauc’s formula. The hydrodynamic size (d.nm), PDI, and the ζ-potential (mV) of the fresh NCs were determined by dilution of each of the NCs suspension in deionized water and pH at 6, 7, and 8. These analyses were performed in DLS using a Zetasizer (Nano ZS Malvern apparatus VASCO Flex™ Particle Size Analyzer NanoQ V2.5.4.0) at 25 °C.

#### 4.4.1. Encapsulation Efficiency and Loading Capacity

The Loading Capacity (LC, %) and the Encapsulation/entrapped Efficiency (EE, %) were carried out to evaluate the compatibility of the selected active ingredient (geraniol) as well as to ensure an efficient encapsulation of the geraniol in ZnOGer1 NCs, ZnOGer2 NCs, and ZnOGer3 NCs. The following formulas were used to determine EE and LC [59]:$$\mathrm{EE}\;\left(\%\right)=\frac{\mathrm{amount}\;\mathrm{of}\;\mathrm{geraniol}\;\mathrm{in}\;\mathrm{the}\;\mathrm{capsules}\;}{\mathrm{total}\;\mathrm{amount}\;\mathrm{of}\;\mathrm{geraniol}\;\mathrm{initially}\;\mathrm{added}\;}\times 100\%$$
$$\mathrm{LC}\;\left(\%\right)=\frac{\mathrm{amount}\;\mathrm{of}\;\mathrm{geraniol}\;\mathrm{in}\;\mathrm{the}\;\mathrm{capsules}\;}{\mathrm{total}\;\mathrm{capsules}\;\mathrm{weight}}\times 100\%$$
where the amount of geraniol in the micelles was quantified spectrophotometrically. All data were expressed as the mean value of three independent batches of the ZnOGer1 NCs, ZnOGer2 NCs, and ZnOGer3 NCs.

#### 4.4.2. Geraniol Release Studies from ZnO-Based NCs

Geraniol release studies from ZnO-based NCs were conducted to assess the effects of active ingredient and manufacturing processes on geraniol release during formulation development and as quality control to support the batch release. The in vitro geraniol release profile from ZnOGer1 NCs and ZnOGer2 NCs was performed using a dialysis membrane (Pur-A-Lyzer Midi 1000) under different temperature conditions: (i) 25 ± 0.5 °C and (ii) 35 ± 0.5 °C. Briefly, 1 mL of each micellar formulation was introduced into a dialysis bag placed in 40 mL of phosphate buffer solution (PBS; pH = 7.2), and kept at a consistent temperature and stirring rate over 96 h. Sample aliquots (2 mL) were withdrawn periodically (0, 1, 2, 3, 4, 8, 16, 24, 48, 72, and 96 h) and replaced with equal volumes of fresh dissolution medium. NCs were suitably diluted and analyzed by the UV–Vis, and drug release was estimated. The release profile of geraniol from NCs was interpreted with four mathematical models: (i) Zero-order, (ii) First-order, (iii) Higuchi, and (iv) the Korsmeyer–Peppas model [60,61,62]. For each measurement, three independent replications were performed.

### 4.5. In Vitro Antifungal Activity against B. cinerea

*B. cinerea* strain B05 was obtained from the culture collection of the Laboratory of Plant Pathology, School of Agriculture, Faculty of Agriculture, Forestry and Natural Environment at Aristotle University of Thessaloniki, and routinely kept on potato dextrose agar (PDA, BD Difco) plates at 25 °C. The procedure was carried out according to Kamou et al. (2022) with minor modifications [9]. ZnOGer1 NCs and ZnOGer2 NCs were added to sterilized PDA medium when they reached 40 °C to obtain 9 different concentrations (31.25, 62.5, 125, 150, 175, 200, 225, 250, and 500 μg/mL), and the pathogen was added after solidification. The growth of the pathogen was measured daily and the SDS was also tested against *B. cinerea*. The growth inhibition of *B. cinerea* was observed and EC_50_ values (half-maximal effective concentration causing 50% inhibition of mycelial growth) were determined with graded dose-response curves. Each treatment had five technical replications, and the experiment was repeated twice.

### 4.6. In Planta Experiments on Cucumbers and Tomatoes

Synthesized ZnOGer1 NCs and ZnOGer2 NCs were sprayed on cucumber and tomato plant leaves previously inoculated with *B. cinerea* in order to investigate their ability to inhibit its growth. Cucumber plants cv. Bamboo in the second leaf stage and tomato plants cv. ACE55 in the fourth leaf stage were inoculated using 10 μL droplets of conidial suspension (2 × 10^6^ sp/mL). The application of the pathogen was performed on the upper leaf part, and 0.1% Tween 20 was added to the spraying solution (water: PDB (Potato Dextrose Broth), 1:1) as a surfactant to maintain the droplet intact. Briefly, plants were sprayed with NCs at concentrations causing 100% inhibition of *B. cinerea* based on the in vitro bioassay. The spraying volume was equal to 5 mL per plant in order to achieve foul coverage of the foliage. Luna^®^ Sensation SC Fungicide was added at the highest recommended dose for vegetables, equal to 30 mL/str (max) with 75 L injection liquid/acre, according to the label, adjusted to our final spraying volume. Plants solely inoculated with the pathogen were used as positive controls. Chemical control plants were treated with conventional fungicide Luna Sensation SC at the highest recommended dose. Each treatment consisted of 8 plants, and the experiment was carried out thrice. A disease index (DI) was used based on symptom development [9] to assess the severity of the disease caused by *B. cinerea* on the leaves of cucumber and tomato.

### 4.7. Statistical Analysis

The EC_50_ values of the NCs against *B. cinerea* in the in vitro bioassays were calculated using a non-linear dose-response curve and 10 replicates per NCs concentration (5 per concentration and repeated twice) by Origin Pro 8 (Data Analysis and Graphing Software). The *in planta* experiments were analyzed by analysis of variance (ANOVA) based on the completely randomized design (CRD), and mean values were computed from the respective replicates. The statistical analyses were performed by one-way analysis of variance followed by Tukey’s post hoc test (*p* ≤ 0.05), which was conducted using SPSS v 25.0 software (SPSS Inc. Chicago, IL, USA).

## 5. Conclusions

The extensive use of synthetic fungicides causes serious concerns for human health and the environment. To overcome these issues, research focuses on alternative environmentally-friendly natural bioactive substances that combine effectiveness with low phytotoxicity to control important fungal pathogens. Nanomaterials and bioactive compounds form nanostructures with superior desirable properties and have the potential to offer alternative solutions. In this term, current work focuses on the eco-friendly synthesis of hydrophobic ZnO@OAm NRs and their modification through SDS as nanocapsules combining NRs and geraniol in different ratios. Encapsulation of nano rods of ZnO@OAm NRs (18 nm) with SDS found with irregular edges. The encapsulation of geraniol was found to be efficient, and the release profile of geraniol from ZnOGer1 NCs and ZnOGer2 NCs presented thermal stability and sustainable-release tendency between 25 °C and 35 °C over 96 h. The antifungal activity of ZnOGer1 NCs and ZnOGer2 NCs was evident in both in vitro and *in planta* experiments. Τhe EC_50_ values of ZnOGer1 NCs, ZnOGer2 NCs, and ZnO NCs against *B. cinerea* were calculated at 176 μg/mL, 150 μg/mL, and >500 μg/mL, respectively. Furthermore, from the *in planta* study, ZnOGer1 NCs and ZnOGer2 NCs showed slightly higher inhibition of *B. cinerea* than chemical Luna Sensation SC on cucumber plants after 96 h. No phytotoxic effect was observed after NCs-treatment on both tomato and cucumber plants. Concluding, nanoemulsions and nanocapsules combining inorganic-based nanoparticles and/or EOs with the different release of active ingredients could have a curative and protective effect on plants and also deliver micronutrients to agricultural crops for an increasing yield.

## Figures and Tables

**Figure 1 plants-12-01074-f001:**
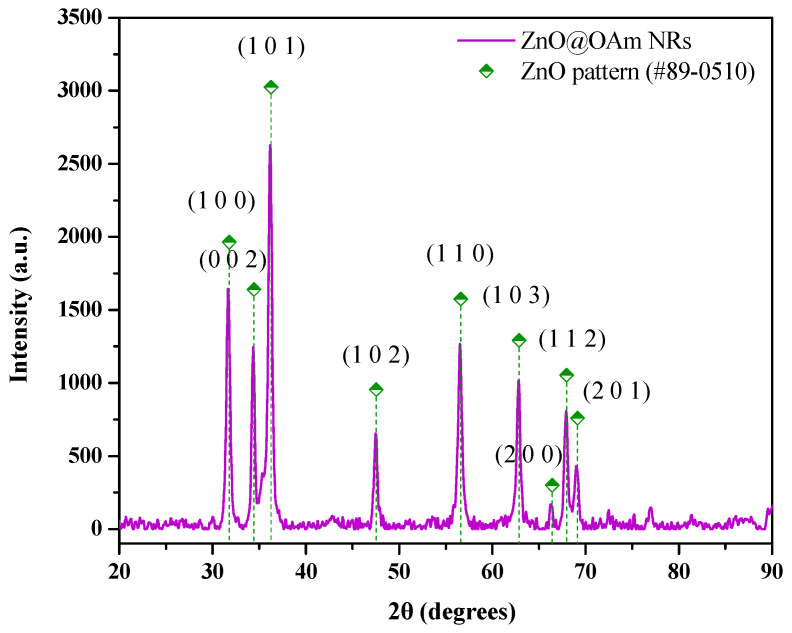
X-ray diffraction (XRD) pattern of the synthesized ZnO@OAm NRs with the main diffraction peaks of ZnO.

**Figure 2 plants-12-01074-f002:**
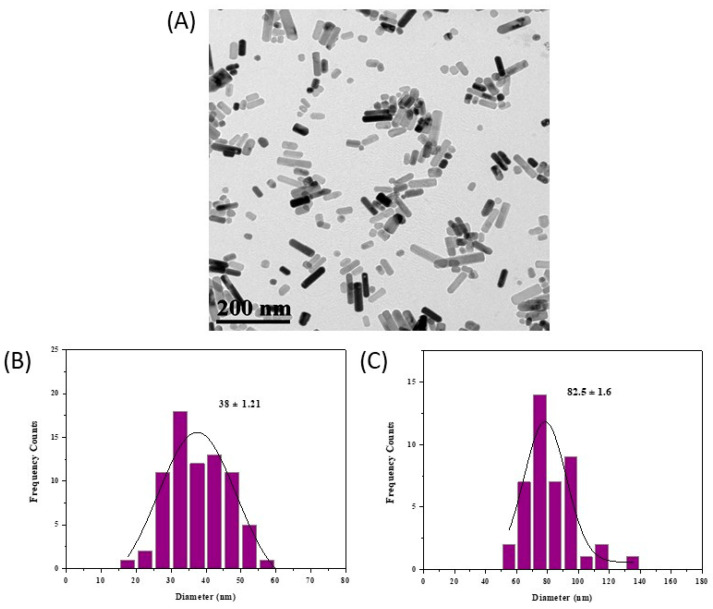
TEM image of ZnO@OAm NRs (NRs; rod-shaped nanoparticles) (**A**), average length histogram and Gaussian fit (38 ± 1.21 nm); and (**B**) average length histogram and Gaussian fit (82.5 ± 1.6 nm) (**C**) of the NRs. The average size of particles is expressed as mean size ± SD.

**Figure 3 plants-12-01074-f003:**
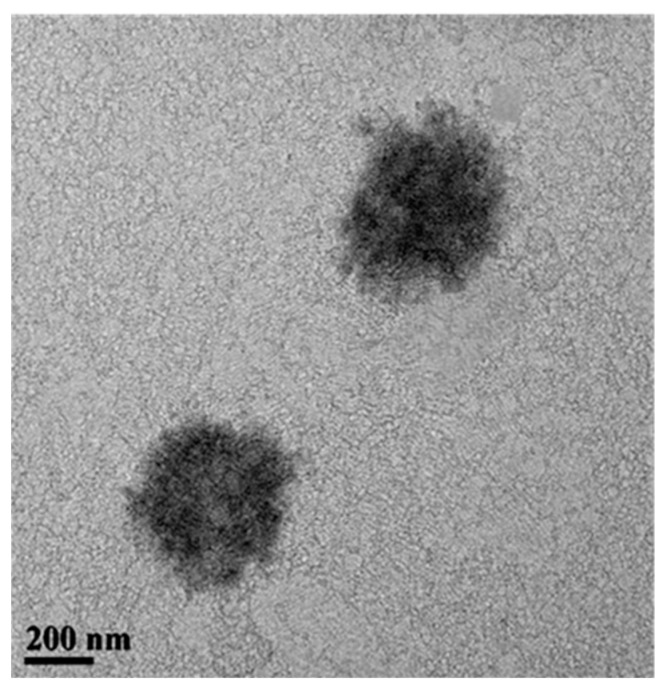
TEM image of the ZnO NCs presents ZnO@OAm NRs in form of nanocapsule.

**Figure 4 plants-12-01074-f004:**
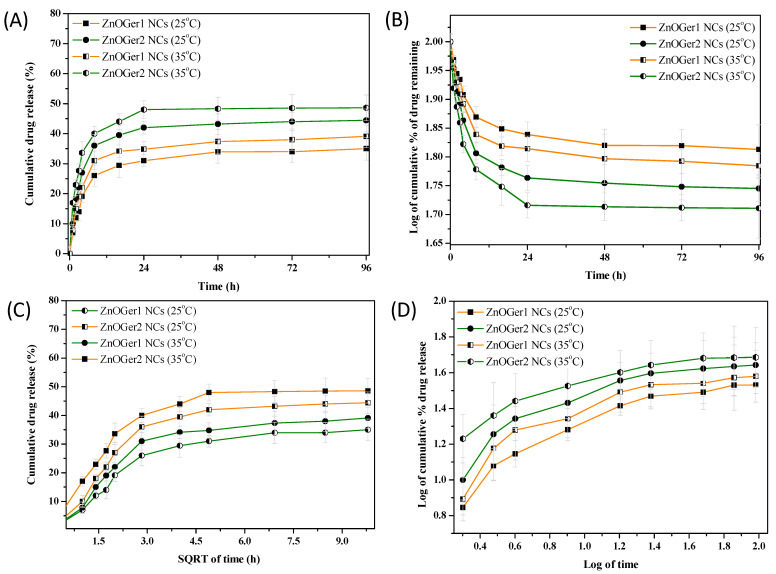
Effects of temperature (25 ± 0.5 °C and 35 ± 0.5 °C) on the stability of ZnOGer1 NCs and ZnOGer2 NCs released geraniol in a 96 h period. Pharmacokinetics for analyzing the release mechanism via zero-order (**A**), first-order (**B**), Higuchi (**C**), and Korsmeyer–Peppas models (**D**).

**Figure 5 plants-12-01074-f005:**
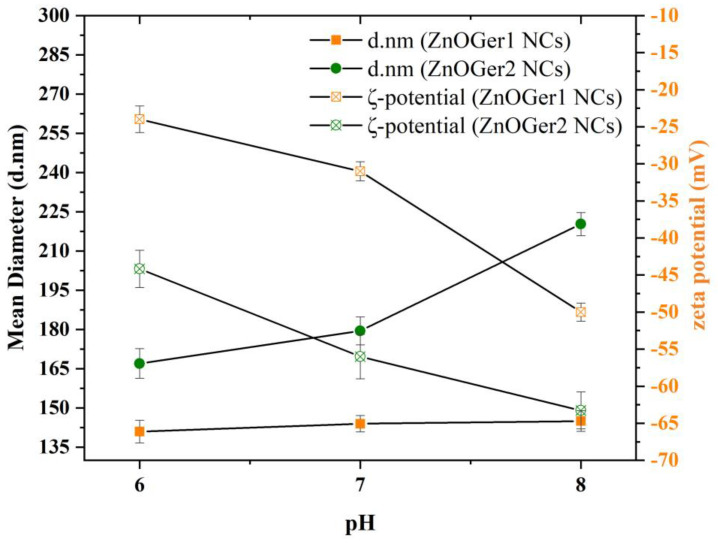
Physicochemical parameters and stability evaluation ZnOGer1 NCs and ZnOGer2 NCs containing commercial geraniol. The stability evaluation was performed for pH values 6, 7, and 8. Mean diameter (d.nm) by dynamic light scattering (DLS), ζ-potential (mV), and pH. All analyses were performed in triplicate (n = 3; Mean ± SD) at 25 °C.

**Figure 6 plants-12-01074-f006:**
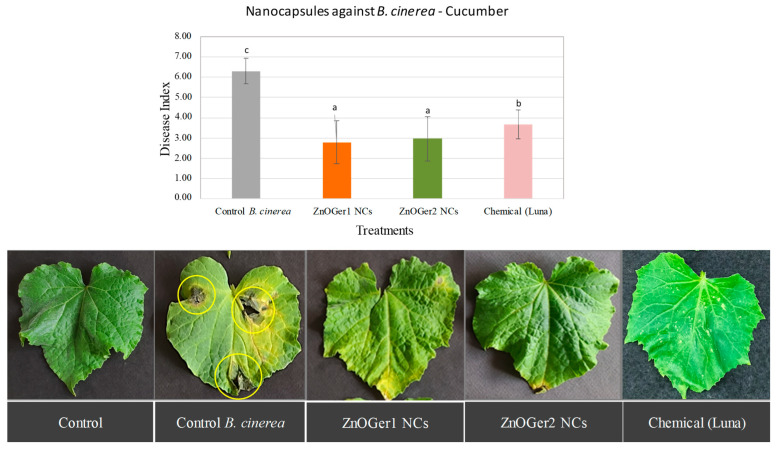
Effect of ZnOGer1 NCs and ZnOGer2 NCs on the severity of disease caused by *B. cinerea* on cucumber leaves in pots under controlled conditions. *B. cinerea* control plants were inoculated only with the pathogen (control *B. cinerea*). The disease was assessed according to the disease index (DI) [9] after 96 h. The experiment was repeated three times. Different letters indicate significant differences according to Tukey’s test at *p* ≤ 0.05. Error bars represent standard deviation.

**Figure 7 plants-12-01074-f007:**
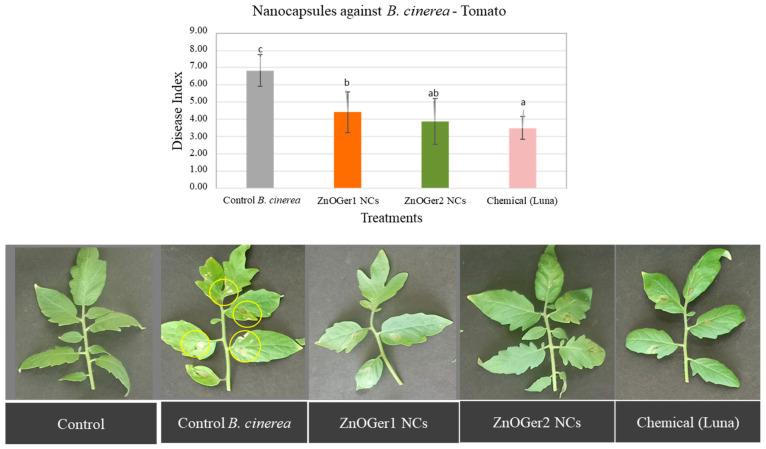
Effect of ZnOGer1 NCs and ZnOGer2 NCs on the severity of disease caused by *B. cinerea* on tomato leaves in pots under controlled conditions. *B. cinerea* control plants were inoculated only with the pathogen (control *B. cinerea*). The disease was assessed according to the disease index (DI) [9] after 96 h. The experiment was repeated three times. Different letters indicate significant differences according to Tukey’s test at *p* ≤ 0.05. Error bars represent standard deviation.

**Table 1 plants-12-01074-t001:** Hydrodynamic size (z-average; d.nm), ζ-potential (mV), and polydispersity (PDI) values of ZnOGer1 NCs, ZnOGer2 NCs, and ZnO NCs at pH values 6, 7, and 8. The analyses were performed in triplicate (n = 3; mean value ± SD).

Nanoemulsions	pH Value	Z-Average (d.nm) ± SD	PDI ± SD	z Potential (mV) ± SD
ZnOGer1 NCs	8	145 ± 2.91	0.37 ± 0.04	−50 ± 1.23
	7	144 ± 1.51	0.31 ± 0.08	−31 ± 1.29
	6	141 ± 4.33	0.25 ± 0.03	−24 ± 0.27
ZnOGer2 NCs	8	220 ± 2.43	0.42 ± 0.01	−63 ± 1.55
	7	180 ± 5.31	0.39 ± 0.08	−56 ± 3.00
	6	167 ± 5.69	0.34 ± 0.02	−44 ± 1.55
ZnO NCs	8	256 ± 2.27	0.48 ± 0.03	−68 ± 2.42
	7	234 ± 2.35	0.45 ± 0.06	−53 ± 3.16
	6	222 ± 3.51	0.43 ± 0.04	−51 ± 2.03

Mean ± SD: mean value of three independent batches of the NCs ± Standard deviation.

## Data Availability

Data is contained within the article or Appendix A.

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
