# Peer review of "Nanocapsules of ZnO Nanorods and Geraniol as a Novel Mean for the Effective Control of Botrytis cinerea in Tomato and Cucumber Plants"

_plants, 2023, doi:10.3390/plants12051074_

Round 1

Reviewer 1 Report

The presented paper demonstrates some novel approaches to control Botrytis infections in plants. The authors presented the results in a clear way and conclusions follow the presented results. 

Author Response

Our response: We would like to thank the Reviewer for the comments and suggestions. 

Reviewer 2 Report

The MS submitted by Tryfon et al., on the "Nanocapsules of ZnO nanorods and geraniol as novel mean for effective control of Botrytis cinerea in tomato and cucumber plants" organized and written well. Presentation of the MS was good. However, the following issues to addressed before it get published.

Comments:

1. Major issue with this MS is the figures are not properly cited and fixed. Many figures are missing in the MS.

2. In introduction, Line 111- 125 can be concise and it looks like a method description.

3. Line no. 36: can be reframed

4. Summary can be improved

5. Reframe the Line no. 447

6. If the expansion provided first in the MS (Ex. TEM, XRD, TGA, Escherichia, etc.) no need to give full form again.

7. Italics should be provided to all the organisms name (Ex. B. cinerea)

Author Response

Comments:

  1. Major issue with this MS is the figures are not properly cited and fixed. Many figures are missing in the MS.

Our response: It has been corrected.

  1. In introduction, Line 111- 125 can be concise and it looks like a method description.

Our response: It changed accordingly.

  1. Line no. 36: can be reframed

Our response: The sentence in Line 36 reframed as “None of the treatments caused phytotoxic effects”.

  1. Summary can be improved

Our response: Conclusions have changed accordingly.

  1. Reframe the Line no. 447

Our response: The sentence was reframed as “Reaction yield was calculated at 38% based on the metal precursor”.

  1. If the expansion provided first in the MS (Ex. TEM, XRD, TGA, Escherichia, etc.) no need to give full form again.

Our response: Changed.

  1. Italics should be provided to all the organisms name (Ex. B. cinerea)

Our response: Changed.

Reviewer 3 Report

-In vitro antifungal activity:

The antifungal activity of the as-prepared ZnO NCs, ZnOGer1 NCs, and ZnOGer2 234 NCs. The experiment should have been conducted using pure geraniol to determine its activity against the fungus and comparing its activity with the NCs. Its result is not included in Fig S4, as well in the  In planta antifungal activity. Moreover, the applied concentrations of ZnOGer1 NCs, ,ZnOGer2 NCs and Luna SC have to be determined

-what is the volume and final concentration of the applied NCs and Luna SC during in planta antifungal activity?

-Line 217, 218: you reported “there is not any significant changes of the hydrodynamic sizes of the NCs in the studied pH values”, while results in table 1 showed varying sizes for ZnOGer2 NCs corresponding to different pH values

-The numbering of the headings and subheadings is wrong, please check

-Many photos are missing: Figure 2 (A, B and C), Figure 3 (A,B) and Figure 4 (A, B, C and D), Figure 6 and Figure 7 are  missed. Please add the missing figures and re-number the figures correctly

-Line 172: Figure 3 refers to FT−IR and UV−Vis spectrum, not for TEM images as written in lines 145, 160

- Line 290: “Figure S1”, in the supplementary data Fig S1 assigned for Thermogravimetric analysis not for rod-like nanostructures

- Line 200, 204: correct “Table S1” to be “Table S2”

Line 37, 185: write “B. cinerea” in italic

Line 441: give number for the reference “de Oliveira et al., 2020” and write the reference details in the list

The attached file may be useful

Author Response

Reviewer 3 (Round 1)

Comments and Suggestions for Authors

  1. In vitro antifungal activity:

The antifungal activity of the as-prepared ZnO NCs, ZnOGer1 NCs, and ZnOGer2 234 NCs. The experiment should have been conducted using pure geraniol to determine its activity against the fungus and comparing its activity with the NCs. Its result is not included in Fig S4, as well in the In planta antifungal activity. Moreover, the applied concentrations of ZnOGer1 NCs, ZnOGer2 NCs and Luna SC have to be determined

The EC50 values of ZnOGer1 NCs, ZnOGer2 NCs, and pure geraniol against B. cinerea were calculated at 176 µg/mL, 150 µg/mL, and 235 µg/mL, respectively. The applied concentration was equal to the double of the EC50 value in order to achieve a lethal dose. Pure geraniol was tested in our previous study (Kamou et al 2022). A phrase was added in Lines 384, 385“… while the EC50 value of pure geraniol has been found in our previous study equals to 235 µg/mL[9].

  1. what is the volume and final concentration of the applied NCs and Luna SC during in planta antifungal activity?

The spraying volume was equal to 5 mL per plant in order to achieve foul coverage of the foliage. Luna® Sensation SC Fungicide (Bayer Crop Science, fluopyram 250 g/L and trifloxystrobin 250 g/L) was added at the highest recommended dose for vegetables: 30 ml/str (max) with 75 L injection liquid/acre, according to the label, adjusted to our final spraying volume. A sentence was added in the Lines 555-559. 

  1. Line 217, 218: you reported “there is not any significant changes of the hydrodynamic sizes of the NCs in the studied pH values”, while results in table 1 showed varying sizes for ZnOGer2 NCs corresponding to different pH values

Our response: The sentence in Lines 233, 234 has been changed as “Sizes of ZnOGer2 NCs are varied corresponding to different pH values”.

  1. The numbering of the headings and subheadings is wrong, please check

Our response: The numbering of the headings and subheadings has been corrected.

  1. Many photos are missing: Figure 2 (A, B and C), Figure 3 (A,B) and Figure 4 (A, B, C and D), Figure 6 and Figure 7 are  missed. Please add the missing figures and re-number the figures correctly

Our response: Figures 2, Figure 3, Figure 4, Figure 6, and Figure 7 have been added in the manuscript despite there was in the submitted manuscript file. We think that it was a problem from the system during uploading the file.

  1. Line 172: Figure 3 refers to FT−IR and UV−Vis spectrum, not for TEM images as written in lines 145, 160

Our response: Changed.

  1. Line 290: “Figure S1”, in the supplementary data Fig S1 assigned for Thermogravimetric analysis not for rod-like nanostructures

Our response: It has been corrected.

  1. Line 200, 204: correct “Table S1” to be “Table S2

Our response: Changed.

  1. Line 37, 185: write “B. cinerea” in italic

Our response: It has been changed.

  1. Line 441: give number for the reference “de Oliveira et al., 2020” and write the reference details in the list

Our response: The reference “de Oliveira et al., 2020” has the number [59] and has been added in the reference list.

Reviewer 4 Report

The manuscript is well written, though the minor issues need to be resolved.

Change the keywords with novel words not used in the abstract.

Check the typo errors throughout the paper.

Conclusion needs to be added with future perspectives of the research.

Try to add more recent references related to the research.

Author Response

Reviewer 4 (Round 1)

Comments and Suggestions for Authors

The manuscript is well written, though the minor issues need to be resolved.

  1. Change the keywords with novel words not used in the abstract.

Our response: Keywords were changed as “nanoformulations; nanofungicides; essential oil encapsulation; antifungal activity; geraniol control release; plant protection; ”.

  1. Check the typo errors throughout the paper.

Our response: Checked.

  1. Conclusion needs to be added with future perspectives of the research.

Our response: It changed accordingly.

  1. Try to add more recent references related to the research.

Our response: The literature about the current study is limited, but we tried to conclude the most of the relevant recent references.

Round 2

Reviewer 2 Report

Authors fulfilled the all the comments raised by me and hence it can be accepted for publications

Author Response

The authors thank the reviwer  for the time to review the MS

Reviewer 3 Report

Many thanks to the authors for responding to comments Please, through the statistical results, please check this note:

-Line 217, 218: you reported “there is not any significant changes of the hydrodynamic sizes of the NCs in the studied pH values”, while results in table 1 showed varying sizes for ZnOGer2 NCs corresponding to different pH values

Author Response

The phrase has changed

The authors thank the reviewer for the time to revew the MS
